# Diel population and functional synchrony of microbial communities on coral reefs

Linda Wegley Kelly [1], Craig E. Nelson [2], Andreas F. Haas[3], Douglas S. Naliboff[1], Sandi Calhoun[1], Craig A. Carlson[4], Robert A. Edwards[1], Michael D. Fox [5], Mark Hatay[1], Maggie D. Johnson[5,6], Emily L.A. Kelly[5], Yan Wei Lim[1], Saichetana Macherla[1], Zachary A. Quinlan [2], Genivaldo Gueiros Z. Silva[1], Mark J.A. Vermeij[7,8], Brian Zgliczynski[5], Stuart A. Sandin[5], Jennifer E. Smith[5] & Forest Rohwer[1,9]

On coral reefs, microorganisms are essential for recycling nutrients to primary producers through the remineralization of benthic-derived organic matter. Diel investigations of reef processes are required to holistically understand the functional roles of microbial players in these ecosystems. Here we report a metagenomic analysis characterizing microbial communities in the water column overlying 16 remote forereef sites over a diel cycle. Our results show that microbial community composition is more dissimilar between day and night samples collected from the same site than between day or night samples collected across geographically distant reefs. Diel community differentiation is largely driven by the flux of *Psychrobacter* sp., which is two-orders of magnitude more abundant during the day. Nighttime communities are enriched with species of *Roseobacter*, *Halomonas*, and *Alteromonas* encoding a greater variety of pathways for carbohydrate catabolism, further illustrating temporal patterns of energetic provisioning between different marine microbes. Dynamic diel fluctuations of microbial populations could also support the efficient trophic transfer of energy posited in coral reef food webs.

[1] Department of Biology, San Diego State University, San Diego 92182, USA. [2] Department of Oceanography and Sea Grant College Program, Center for Microbial Oceanography: Research and Education, University of Hawai'i at Mānoa, Honolulu 96822, USA. [3] Department of Marine Microbiology and Biogeochemistry, NIOZ Royal Netherlands Institute for Sea Research and Utrecht University, Texel, The Netherlands. [4] Department of Ecology, Evolution and Marine Biology, University of California, Santa Barbara 93106, USA. [5] Scripps Institution of Oceanography, University of California, San Diego 92093, USA. [6] Smithsonian Marine Station, Ft. Pierce, FL 34949, USA. [7] Caribbean Marine Biological Institute (CARMABI), Willemstad, Curaçao. [8] Aquatic Microbiology, University of Amsterdam, Amsterdam, The Netherlands. [9] Viral Information Institute, San Diego State University, San Diego 92182, USA. These authors contributed equally: Linda Wegley Kelly, Craig E. Nelson, Andreas F. Haas. Correspondence and requests for materials should be addressed to L.W.K. (email: lwegley@gmail.com)

Microorganisms comprise the majority of biomass in the oceans and their role in the decomposition of organic substrates is critical to nutrient cycling, as well as for channeling nutrients and energy to higher trophic levels[1–3]. The water column overlying tropical coral reefs comprises a complex mixture of oligotrophic offshore waters and reef water enriched with the organic carbon and nitrogen substrates exuded by the benthic community[4,5], which establishes the base of a robust microbial food web[6,7]. Coral reef benthic communities can directly consume bacterioplankton from the overlying water column via suspension feeding[8–11] thereby reducing energy loss that would otherwise be respired by microbes. These top down consumption processes reduce microbial production in the water column and simultaneously enhance transfer of microbial biomass to metazoan consumers in the benthos. The mechanisms sustaining these retention processes, yet poorly understood, promote the tight recycling of materials and production of high consumer biomass observed in coral reef ecosystems[12].

A key question in reef ecology is identifying how benthic communities influence reef microbial community structure and function. Microbial biomass and community structure on reefs are linked to local conditions, such as the composition of benthic assemblages[13,14] and allochthonous inputs[15,16]. As reefs shift toward algal-dominated states microbial production increases and becomes a greater energetic sink compared to that observed in a coral dominated system[17,18]. Diel investigations on reefs are required to better understand the heterotrophic metabolisms that dominate at night, including fundamental processes influenced by microbial communities, such as reef dissolution and boundary layer anoxia[19–21]. Furthermore, the influence of rhythmic growth patterns versus predation on reef microbial community structure remains virtually unknown.

The majority of studies on coral reefs have been conducted during the day when diurnal processes predominately associated with corals and benthic algae exhibit highest rates of primary production and calcification. The hazards of apex predator feeding behaviors and of navigating boats through the reef during the night complicate the logistics of acquiring field diel measurements and nocturnal sample collections outside of controlled but artificial environments such as aquaria. Despite these challenges, few studies have shown that the dark reef (i.e., the matrix of crevices and caves) is a hotspot for microorganisms[22] and meiofauna (e.g., amphipods and other tiny invertebrates) that are active during the night[23]. Autochthonous copepods have been observed moving into the water column at night to feast on the planktonic communities[24,25]. Remote cameras and hydrophones have also recorded nocturnal migrations of invertebrates from the dark reef to the benthic surfaces[23]. To further distinguish trophic linkages between macroorganisms and microorganisms, and provide a more holistic understanding of the structure and function of coral reef communities, measurements of microbial dynamics in natural reefs over a complete diel cycle are needed.

The study described here used a novel apparatus to collect diel biochemical and metagenomic samples from remote coral reefs to characterize microbial community dynamics in the water column overlying 16 forereef plots over a 24 h period. Metagenomic characterizations of environmental microbes can reveal linkages between spatial and temporal community dynamics, biogeochemical fluxes, and ecological niche partitioning (e.g. refs. [26–28]). Our results illustrate a dramatic and consistent shift in day versus night microbial communities, which is reflected both in the taxonomic structure and the metabolic capacity encoded by the populations. This study highlights ecosystem functions on reefs that support dynamic fluctuations of diel microbial populations, capturing a key aspect of microbial ecology implicated in promoting trophic transfer of energetic resources through the microbial food web in tropical reef ecosystems.

## Results

**Reef sites and day–night sampling.** The data presented here were collected on a cruise to the southern Line Islands located in the Republic of Kiribati (Central Pacific) from October 22 through November 6, 2013. The study site encompasses three uninhabited coral islands and one atoll: Vostok (−10.0609, −152.309), Millennium (−9.95080, −150.215), Starbuck (−5.62891, −155.925), and Malden (−4.01407, −154.973) separated by 800 km of latitudinal distance and exhibiting variance in ocean productivity and nutrient regimes between islands[29]. On each island, seawater overlying the benthos (<0.5 m from the bottom) was collected from four distinct forereef sites over a 24-h period ($N = 16$). Benthic chambers were constructed over each reef plot ($0.75\ m^2$) to provide structure for autonomous sampling at night (Fig. 1a). The forereef sites were subjected to high flushing through the reef matrix, therefore water exchange within the benthic chambers ranged between 2.52 and $5.48\ l\ min^{-1}$ (Supplementary Fig. 1). Each reef plot included a multiparameter sonde for continuous monitoring of temperature, conductivity, dissolved oxygen, and pH (Manta2, Eureka Water Probes, Austin, TX, USA) and was rigged with a time-controlled sampling device to collect reef water during the night.

These remote Southern Line Islands comprise some of the most pristine coral reefs in the world; they are intact with high biomasses of apex predators[30–32] and dominated by calcifying coral and algae[33]. All benthic chambers were placed on forereef sites (10 m depth) which were predominately composed of reef-building macroorganisms including scleractinian corals, crustose coralline algae (CCA) and calcified macroalgae (e.g., *Halimeda* sp.; Fig. 1b, Supplementary Table 1; mean = 90.1%; min = 62.5%; max = 100%). Fleshy macroalgae and turf algae were also present, but at lower percent coverage (Fig. 1b; Supplementary Table 1). Dissolved oxygen changed an average of $38\ \mu moles\ l^{-1}$ (min = $182.7 \pm 2.31$, max = $220.6 \pm 7.61$) and pH ranged from 8.13 to 8.49 over a diel cycle. Temperature differences over the 24 h sampling period were lowest on Vostok and Millennium (0.1 −0.4 °C) and highest on Malden (1 °C) (Fig. 1c). Dissolved organic carbon (DOC) concentrations were higher on the reef ($139.9 \pm 19.1\ \mu moles\ kg^{-1}$) compared to offshore waters ($96.3 \pm 4.9\ \mu moles\ kg^{-1}$). A subset of the DOC pool characterized as humic-like fluorescent dissolved organic matter (fDOM) was also measured in higher concentrations on the forereef and did not differ between open reefs and benthic chambers at the start of incubations ($p > 0.98$). Both DOC and fDOM concentrations demonstrated significantly higher concentrations in the night samples across islands ($p < 0.05$), indicating either enhanced release or reduced consumption of organic materials during the night (Fig. 1d); in all cases concentrations returned to starting levels by the following midday sampling, indicating either removal or dilution.

**Population dynamics of day–night microbial communities.** The microbial community structure during the day (t0 and t24) and the night (t12) is described from four distinct reef sites on each of the four islands: Vostok, Millennium, Starbuck, and Malden ($N = 16$ sites and 48 samples). Microbial community composition differed strongly between day and night across all reef sites and islands (PERMANOVA $p < 0.0001$; Supplementary Fig. 2a). Microbial phylogenetic and functional community composition was more similar across sites and islands during the day (weighted unifrac distance mean 0.24) than between the day and the night on the same island or even within the same benthic

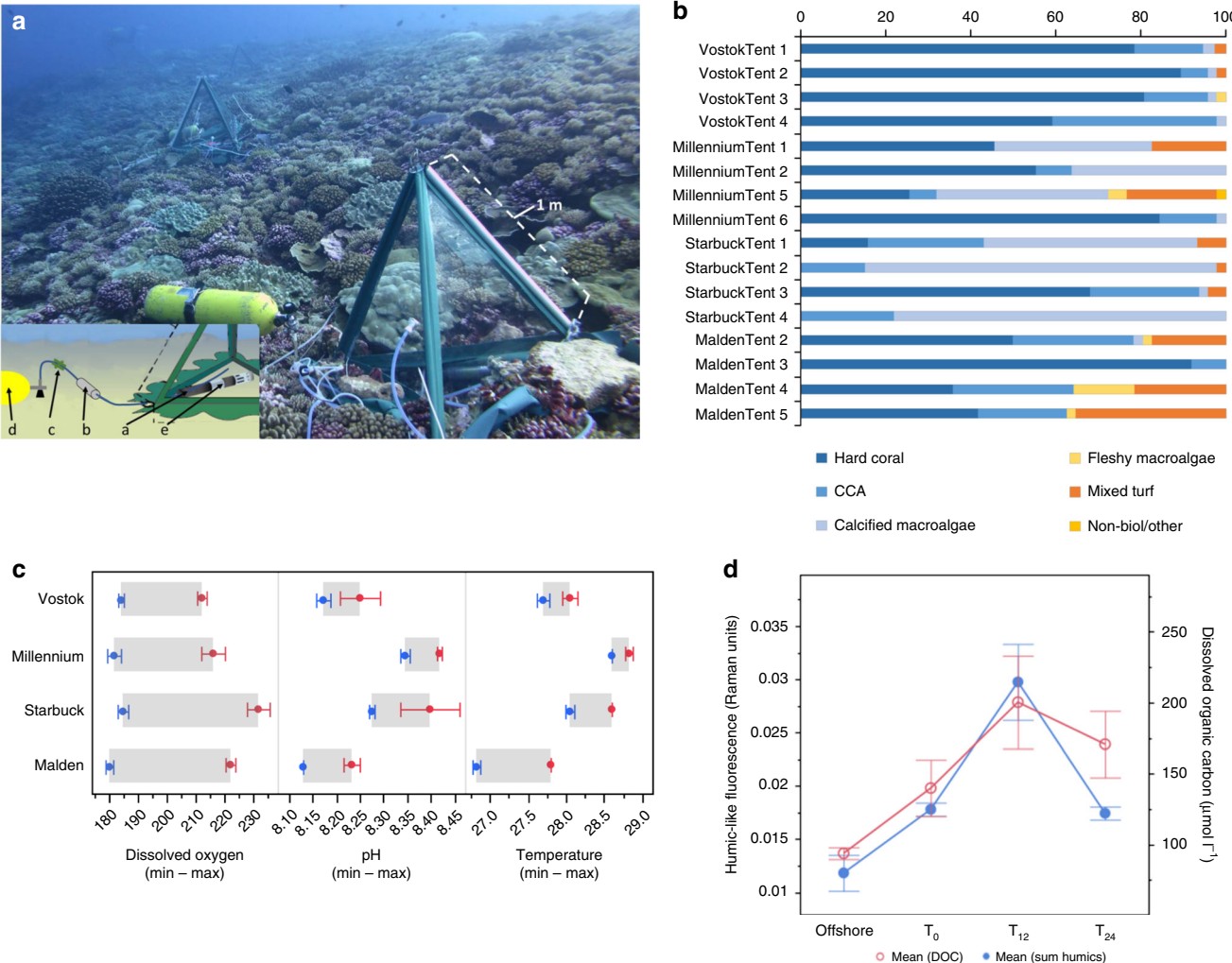

**Fig. 1** Diel sampling design and reef site properties. **a** Collapsible benthic isolation tents (cBIT) were equipped with an autonomous night sampling mechanism. Microbial communities were collected via negative pressure through silicone tubing (a) onto a 0.22 μm Sterivex filter (b). Temporal sampling was triggered by an in-line time-release valve (c) preceding a steel tank that provided vacuum pressure (d). All benthic chambers included a multiparameter sonde (e) to continuously monitor temperature, specific conductivity, dissolved oxygen, and pH. Photo credit: Jennifer Smith. **b** Percent cover of benthic assemblages at each reef site. **c** Geochemical measurements for a 24-h diel period (boxes represent the mean ± SD from all four islands for the minimum (Min) and maximum (Max) of each analyte, $N = 16$); dissolved oxygen (μmoles kg$^{-1}$); Temperature °C. **d** Dissolved organic carbon (DOC) concentrations (right-hand $y$-axis; μmoles l$^{-1}$ and humic-like fluorescent dissolved organic matter (fDOM; left-hand $y$-axis; Raman units) measured in the water column offshore and within each benthic chamber on the forereef at three time-points (points represent the mean ± s.d. from all four islands, $N = 83$)

chambers (weighted unifrac distance means 0.31; $p < 0.0001$; Supplementary Fig. 2b). Night communities were more variable than day communities at sites on the same island (unifrac 0.26 versus 0.22, $p < 0.0001$) and among islands (unifrac 0.29 versus 0.24; $p < 0.0001$; Supplementary Fig. 2b). Microbial communities differed significantly by Island across all samples and within both day and nighttime subsets of the data (PERMANOVA $p < 0.001$), emphasizing the role of island-scale variation in benthic communities (Fig. 1b, Supplementary Table 1) in structuring microbial consortia[14]. Across all islands microbial communities did not differ statistically between ambient (t0) and chamber endpoint (t24) daytime communities (PERMANOVA $p > 0.25$) or between ambient and chamber nighttime (t12) communities (PERMANOVA $p > 0.37$) but both ambient and chamber diel pairwise comparisons were all significantly different (PERMANOVA $p < 0.05$; Supplementary Fig. 2c). There was no significant interaction between Island and Time or between Island and inside versus

outside of chambers (PERMANOVA $p > 0.05$), clarifying that Islands did not differ in these robust patterns. Within each benthic chamber (reef site), microbial community phylogenetic and metagenomic composition differed more between night and day than between the daytime start (t0) and end (t24) of deployments, or between daytime samples among chambers ($p < 0.0001$; Supplementary Fig. 2b), suggesting both that daytime samples after 24 h were representative of the ambient reef and that patterns in community structure were driven by temporal influences, rather than confinement effects of the benthic chambers.

Day communities were dominated by bacterial taxa most closely related to *Psychrobacter* spp. from the Family Moraxellaceae (Gammaproteobacteria). Abundances of Moraxellaceae taxa during the day ranged from 32.8 ± 5.66% on Starbuck (mean ± s.e.m.) to 70.2 ± 3.47% on Vostok (Fig. 2a). *Synechococcus* spp. (Cyanobacteria) were also enriched during the day

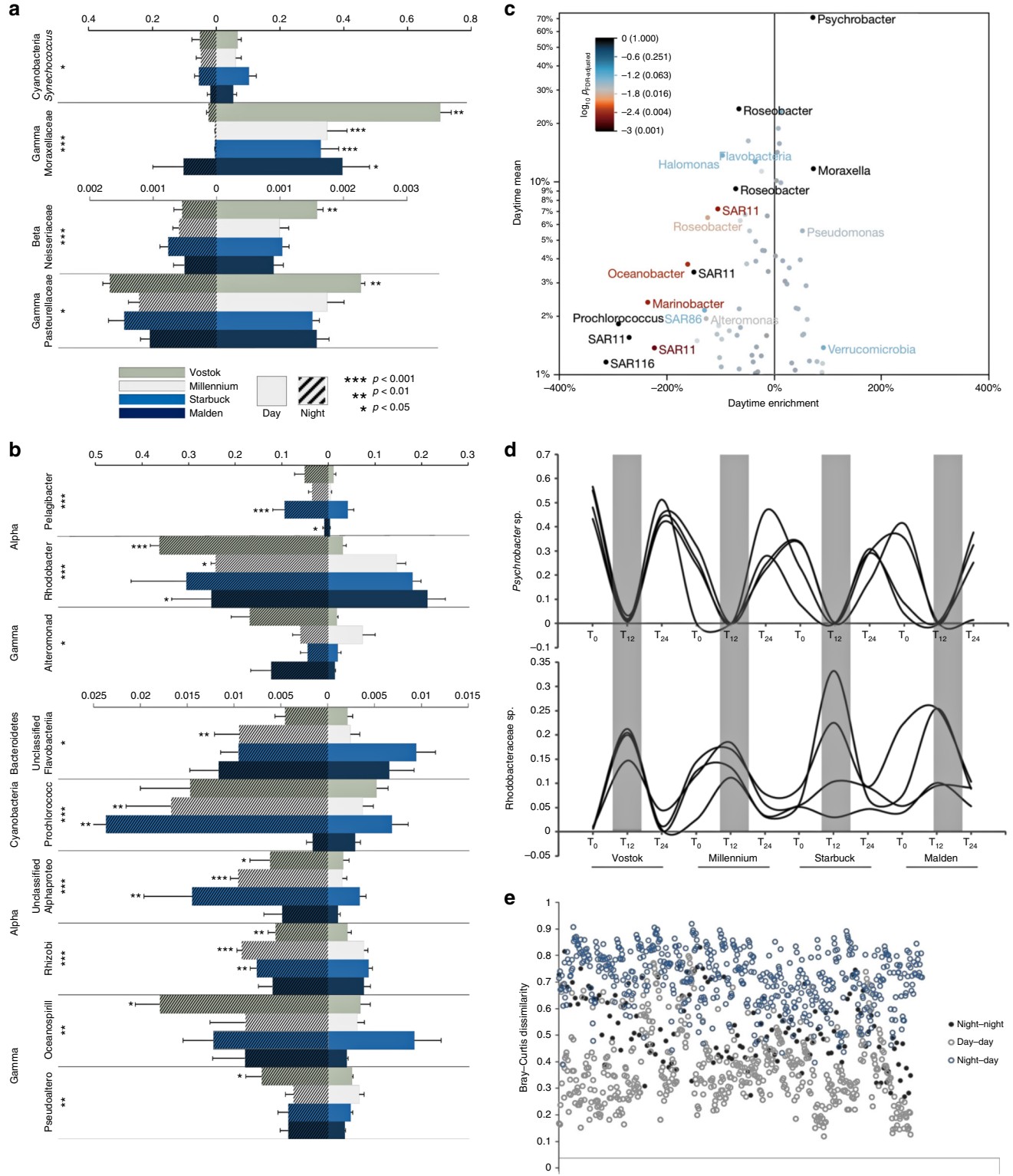

(5.35 ± 1.03–10.3 ± 2.25). Rarer bacterial families significantly enriched during the day included Neisseriaceae (Betaproteobacteria) and Pasteurellaceae (Gammaproteobacteria) (Fig. 2a) and three Families from the Phylum Verrucomicrobia (Akkermansiaceae, Rubritaleaceae, and Verrucomicrobiaceae) (Supplementary Table 2). The most abundant taxa present in the night samples were Alphaproteobacteria belonging to the families Rhodobacteraceae (24.2 ± 0.98–36.2 ± 1.96) and Pelagibacteraceae

(SAR11 Clade, 0.89 ± 0.35–9.41 ± 2.49) and Gammaproteobacteria from the Family Alteromonadaceae (4.49 ± 1.54–16.9 ± 4.05; Fig. 2b). There were several significantly enriched Alphaproteobacteria and Gammaproteobacteria families ranging from 1% to 5% of the community including Rhizobiaceae, Rhodospirillaceae, and Halobacteriovoraceae, and Oceanospirillaceae, Pseudoalteromonadaceae, and Vibrionaceae, respectively (Fig. 2b; Supplementary Table 2). Changes in the relative abundance of microbial

**Fig. 2** Population dynamics between day and night microbial communities. **a** Day-enriched metagenomic taxa are shown in bar plot as the mean proportion of the community (RA relative abundance; x-axis) for each island during the day (right of zero) and during the night (left of zero) separated vertically by common taxa (>1% of total community, upper axis) and rarer taxa (<1%, lower axis). **b** Night-enriched metagenomic taxa are shown as common taxa only (>5%, upper x-axis and 1−5%, lower x-axis). Rarer taxa (<1%) significantly enriched during the night are listed in Supplementary Table 2. Error bars depict s.e.m. (standard error of the mean). Asterisks adjacent to Family designations on the y-axis denote level of significance between day and night samples across all islands ($N = 48$; t-test, FDR adjusted p-value; ***$p < 0.001$, **$p < 0.01$, *$p < 0.05$). **c** Enrichment of day versus night taxa based on metagenomic sequence alignments to the SSU rRNA gene (x-axis). Mean relative abundance of bacterial taxons at day time points (y-axis). Color gradient depicts significance between day and night samples ($N = 48$, FDR-corrected p-value, color gradient log10 scaled). **d** Day/night oscillations of two representative bacterial taxa. Each line represents one replicate reef site at three time points on four islands ($N = 48$). Gray bars depict night time points. y-axis represents the proportion of community. **e** Bray−Curtis dissimilarity in community composition between reef microbial populations. Data points ($N = 1209$) represent a pairwise distance comparison of each sample based on the phylogenetic annotation of SSU rRNA reads extracted from the shotgun sequence libraries. Source data are provided as a Source Data File

phylotypes (16S rRNA genes characterized by alignments to the SILVA database, Supplementary Fig. 3) over a diel cycle on these reefs were substantial (Fig. 2c–e). One taxon within the genus *Psychrobacter* (Family, Moraxellaceae; Phylum, Gammaproteobacteria; Supplementary Fig. 3) could represent up to 56.7% of the microbial population during the day (mean = 36.2%, $N = 32$), but was depleted by nearly two orders of magnitude during the night (mean = 0.5%, $N = 16$) (Fig. 2d). In contrast, several species of Rhodobacteraceae (Alphaproteobacteria) and Halomonadaceae (Gammaproteobacteria) were rarer during the day, but represented the most abundant members of the community at night (Fig. 2c, d).

**Metabolic pathways encoded by day versus night communities.** Microbial community metabolism was annotated against the SEED database (e-value < $10^5$, minimum length of alignment of 45 bp, minimum nucleotide identity of 70%), where each read assigned a gene function was classified into Subsystem Levels 1−3 (metabolic pathways annotated into a stepwise hierarchy)[34]. Hierarchical cluster analysis classified the broadest metabolic designations (Subsystem Level 1, e.g., DNA metabolism) into two groups of predominantly daytime (14 categories) or nighttime enriched metabolism (eight categories) (Fig. 3a). The metabolic pathways encoded by the reef microbes during the day and night followed a similar pattern of community differentiation as the phylogenetic structure where daytime functional potential consistently differed significantly from the night in a dimension orthogonal to the majority of variation among tents and islands (Fig. 3b, Supplementary Fig. 2b). Daytime communities encoded greater abundances of genes for anabolic pathways, such as fatty acids, cofactors, DNA and RNA metabolism, cell wall biosynthesis, molecular regulation and cell cycle pathways (Fig. 3a). Night communities encoded more genes for catabolic pathways (carbohydrate and aromatic compound metabolism) and ATP-dependent functions (membrane transport and motility) (Fig. 3a).

Finer classification of the metabolic pathways (Level 3 subsystems) that were significantly enriched between the day and night communities across all reef sites are depicted in Fig. 3c. Each subsystem was tested using a mixed-effects least-squares linear model for significant differentiation between the fixed effect of day and night within each tent where both tent deployment and island/atoll were random effects; variance estimation was done using restricted maximum likelihood and the false discovery rate was controlled by adjusting p-values according to Benjamini and Hochberg[35]. During the day, several different housekeeping gene pathways were more abundant, including DNA replication and repair, RNA processing and modification, and protein folding. Biosynthetic pathways to build plasma membranes (phospholipids), protein cofactors and Gram negative cell wall components (e.g., folate and outer-membrane proteins, respectively) were all present in higher abundances during the day (Fig. 3c; Supplementary Table 3). Gene pathways for the resistance to antibiotics and heavy metals were also more prevalent in day communities whereby eight unique metabolic pathways (Level 3 subsystems) were significantly enriched (Fig. 3c, symbol R in Virulence panel). The majority of nighttime-enriched pathways were related to catabolic metabolisms including degradation of monosaccharides, disaccharides, & oligosaccharides[18], sugar alcohols[5] and organic acids[4], and purine & pyrimidine catabolism[4]. Several ATP-dependent pathways involved in the transport of organic and nutrient substrates (7 ABC and 1 TRAP transport system), as well as flagellar motility were more abundant at night. Finally, seven distinct pathways to build a cellular capsule were significantly enriched at night (Fig. 3c, symbol C in Cell Wall panel; Supplementary Table 3).

## Discussion

Marine microbes generally demonstrate similar community profiles during the day and night. Previous studies from the North Pacific Subtropical Gyre[36], the English Channel[37], and Monterey Bay[38] all report marginal changes in community structure and gene content between day and night communities. While transcriptional activity of metabolic functions in marine bacterioplankton communities are strongly influenced by diel patterns[36,38,39], significant shifts in composition are more commonly linked to seasonal changes[37,40,41], spatial patterns, such as distance to land[42] and oceanographic processes, such as upwelling[43] and mesoscale eddies[44].

In contrast to open ocean bacterioplankton, our results indicate that the microbial populations inhabiting coral reef waters are subjected to strong diel shifts in relative abundance. For instance, one species of Gammaproteobacteria, most closely related to the Genus *Psychrobacter*, dominated the day community on all reef sites (up to 70% on Vostok), but was depleted to an average of 0.5% at night (Fig. 2c, d). The high prevalence of *Psychrobacter* spp. (Moraxellaceae Family) on these reefs was striking but not unexpected. For comparison, reef microbes collected between 2009 and 2016 from 22 Pacific islands were investigated for the presence of *Psychrobacter*. The abundances of *Psychrobacter* spp. were higher on the reef compared to surface and offshore waters (Supplementary Fig. 4a) and varied widely across reefs ranging from <1% to 40% (Supplementary Fig. 4b). The genus *Psychrobacter* has been characterized as aerobic, non-motile, cold-adapted, and mesophilic[45,46]. Designating the most abundant taxon inhabiting coral reef waters as psychrophilic was surprising. Hence, this discovery that a closely related population of *Psychrobacter* spp. (Supplementary Fig. 3) can dominate tropical marine microbial communities warrants further investigation to better understand the ecology of this clade.

There were also a number of microbial taxa that were common during the night across all islands including one phylotype from

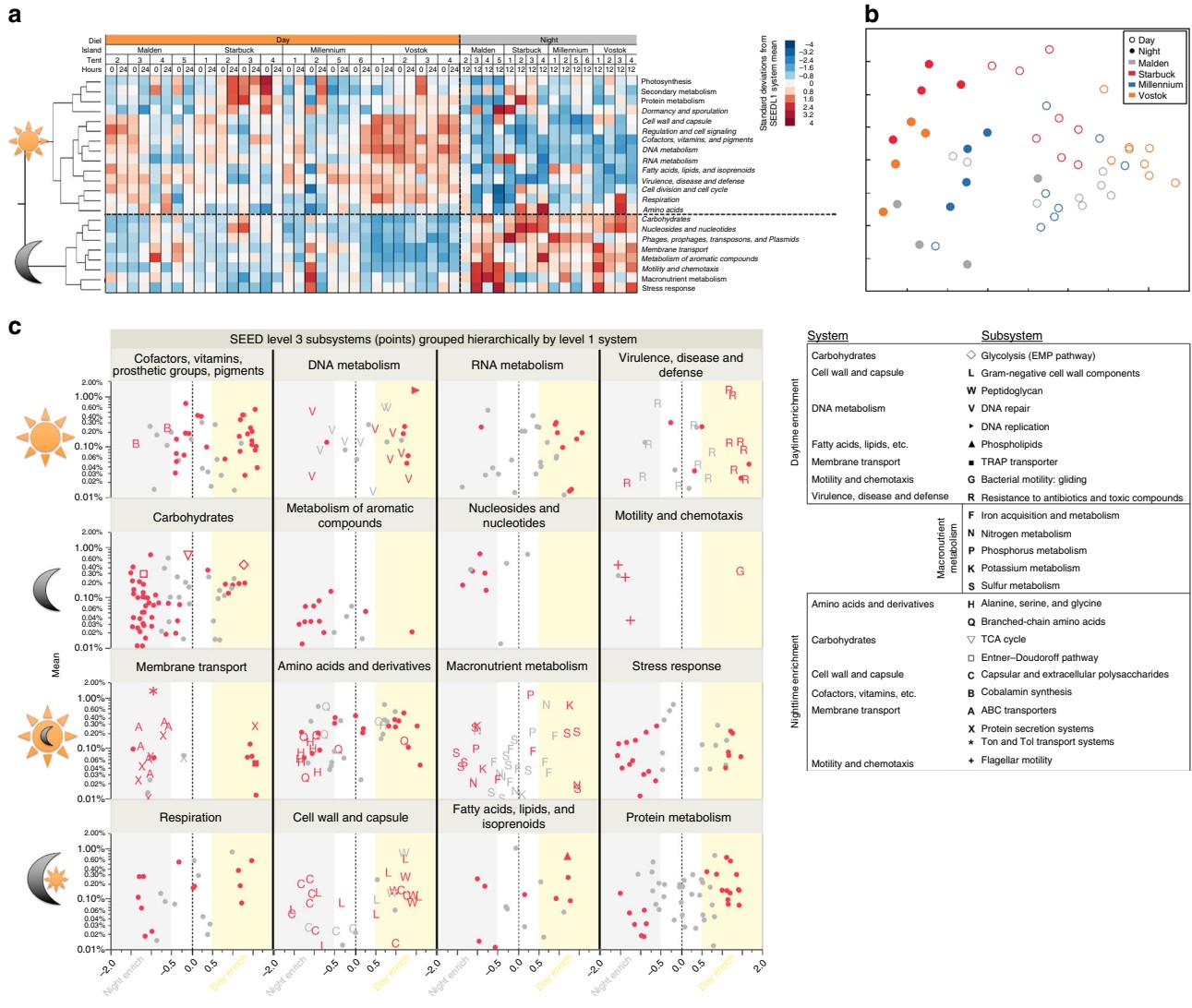

**Fig. 3** Diel shifts in relative abundance of protein coding genes on coral reefs. Hierarchical clustering (Ward's minimum variance method) of broad gene categories (heatmap of SEED Level 1 standard deviations) among samples collected during day and nighttime at each reef site on all four islands (**a**). Multidimensional scaling of sample Bray–Curtis distances calculated from SEED Level 1 relative abundances (**b**). Panel **c** shows relative abundance (*y*; mean percent of sequences per sample) and diel enrichment effect size (*x*; Cohen's *d*, the difference in day and night mean divided by the standard deviation; |*d*| > 0.5 is considered moderately enriched) of protein-coding gene pathways (points, SEED Level 3 Subsystems) colored according to significant difference between day and night at each reef site (red are FDR corrected *p* < 0.05). Top row: cofactors, DNA metabolism, RNA metabolism, and virulence are all enriched during the day. Second row: carbohydrates, metabolism of aromatic compounds, nucleotides, and motility genes are all enriched at night. Third row: membrane transport, amino acids, macronutrient metabolisms, and stress response all tend toward nighttime enrichment; the latter two are not significant at Level 1. Bottom row: respiration, cell wall and capsule, fatty acids and protein metabolism show mixed enrichment of Level 3 pathways. This graphic represents over half of the main subsystems categories in the SEED database; the remaining Level 1 subsystem categories did not contain abundant gene families (Level 3 subsystems mean > 0.1%) with moderate significant enrichment (*p* < 0.05, Cohen's *d* > 0.5), though all Level 1 SEED categories shown in **a** did have at least one Level 3 subsystem significantly different between day and night. Source data are provided as a Source Data File

the family Rhodobacteraceae (Alphaproteobacteria) that on average represented 17% of the night population and up to 33% on Starbuck. Gammaproteobacteria from the Genus *Halomonas* (Oceanospirillales) and *Alteromonas* also represented a significant proportion of the night community (up to 40% at sites on Starbuck and Malden and >50% at sites on Vostok and Malden, respectively). The difference in diel patterns between open ocean communities and those associated with coral reef habitats reflects the potential for a benthic–pelagic coupling of the microbial food web.

The metabolic profile during the day was strongly influenced by the high prevalence of *Psychrobacter* spp. The metabolic profile of genes encoded during the day were predominately anabolic pathways including the biosynthesis of cofactors, vitamins, cell walls, and membranes (lipids), as well as DNA replication and repair. Thus, day community metabolism reflected a strategy for cellular growth, but a limited capacity to catabolize a variety of substrates.

Nighttime communities showed a greater diversity of genes encoding for catabolic functions including a higher abundance of fermentative pathways for energy acquisition. Key pathways in central carbohydrate metabolism also differed between the day and night communities. The more efficient Embden–Meyerhof–Parnas (EMP pathway) was enriched in day communities whereas genes

encoding the Entner–Doudoroff (ED pathway) were significantly more abundant at night, suggesting alternative strategies for catabolism at night[18]. Enriched carbohydrate pathways encoded by the night community also reflect greater availability of sugar alcohols and cycling of C1–C3 compounds that could be metabolized more favorably in low oxygen conditions[47]. Prior research has suggested that temporal cascades of transcriptional activity by different oceanic heterotrophs reflect a mechanism for partitioning energetic resources between members of the microbial community[39,48,49]. These organisms potentially respond to rhythmic metabolic fluctuations through molecular regulation that result in oscillatory patterns or circadian clocks[50]. Coral reef habitats create strong temporal fluxes of biochemicals including oxygen, pH, labile-dissolved organic matter, and inorganic nutrients (e.g. Fig. 1c, d) that could provide essential cues for local biota to evolve oscillatory mechanisms to maximize utilization of resources over a diel cycle.

Coral reef communities appear to maximize efficient processes related to nutrient cycling; dynamic growth and removal processes foster high production in these otherwise resource-limited environments[7,42,51–53]. Temporal synchrony of certain members of the microbial community to exhibit high metabolic rates during the day that correlate with primary productivity may represent another example of such mechanisms. These fast-growing diurnal microbes are diminished at night by (1) reduced growth rates, (2) protist predation that could channel energy into the benthic food web, or (3) viral lysis coordinated to remove a large proportion of the bacterial biomass, providing both reduced activity and available energetic substrates to different members of the community.

The influence of predation versus rhythmic growth patterns on community structure are fundamental questions in microbial ecology that remain largely unanswered. The use of commercial autonomous samplers could provide the capacity to sample different size fractions (i.e., particle associated versus free-living cells), at finer temporal resolution (e.g., hourly) and over consecutive diel cycles. This enhanced time resolution would also provide a means to further characterize dynamics of microbial populations and help resolve the roles of predators, both phage and protists, in community shifts on diel time scales. While it is less logistically feasible to deploy larger sampling equipment on cruises, land-based field studies both on populated islands and remote regions (e.g., Palmyra; the northwest Hawaiian Islands) would provide greater geographic context to these observed diel patterns and allow for comparison across intact and degraded coral reef ecosystems. We hypothesize that the ecological functions driving these microbial fluxes will be diminished as habitats become influenced by anthropogenic perturbations. Further studies are required to better describe the retention of microbes from the water column into the benthic habitat, how this capacity serves to influence production and respiration processes in coral reef ecosystems, and to what degree this functionality is lost as habitats degrade.

## Methods

**Day–night sampling**. Discrete water samples were collected from each reef plot ($N$ = 16) over a 24-h diel cycle: between 0900–1159 on day 1 (t0), 2100−2359 on day 1 (t12), and 0900−1159 on day 2 (t24) for a total of 48 samples. On each of the four islands, samples were collected at one reef site without a benthic chamber at t12 to represent an open nighttime reef community without confinement. Time-zero samples were collected immediately after benthic chambers were deployed and therefore should be representative of the ambient reef water community. Reef water samples (1–3 l) were pumped through 0.22 μm 47 mm polyethersulfone filters (Sterivex, EMD Millipore, Billerica, MA, USA) that were subsequently dried and frozen at −20 °C.

**Sequencing, bioinformatics, and statistics**. DNA extraction and sequence library preparation were completed at San Diego State University (Supplementary

Methods)[54]. For metagenomic comparisons of the microbial community, three time points collected over a diel cycle from four reef sites on each of the four islands were sequenced and analyzed for a total of 48 experimental samples plus four t12 samples collected from open reef sites (depicted in Supplementary Fig. 2). Shotgun libraries (Nextera XT, Illumina, San Diego, CA, USA) were sequenced on the MiSeq II Platform (Illumina, San Diego, CA, USA). Sequence reads were compared to the SEED database[34] for metabolic and taxonomic assignments using SUPERFOCUS, which aligns sequence similarities using RAPSearch2 and performs a 98% clustering of the proteins in the database to reduce computational taxation[55]. Metagenomic reads aligned against the SEED database to get the functional annotation were subsequently extracted to identify the taxa that are encoding the respective protein-coding genes from NCBI using Taxonkit (http://bioinf.shenwei.me/taxonkit/). Shotgun sequence metagenomic libraries generated ~17 million reads with an average length of 225 base pairs after low-quality reads were removed using Prinseq[56]. More than six million significant sequence similarities to the SEED protein database were obtained (Supplementary Table 1). Community dissimilarity was derived from Bray–Curtis distances calculated in R using the Vegan Package[57].

Putative SSU rRNA gene sequences (25,316) were extracted from the shotgun libraries using GenomePeek[58] and aligned to the SILVA v115 SSU database[59]. Alignment, classification, sequence distance calculation, OTU clustering, phylogenetic tree construction, and calculation of among-sample phylogenetic distances were done using the software package mothur[60,61]. The average number of 16S reads per library was 528 (Supplementary Table 1), a lower sequencing depth than amplicon libraries typically generated using next generation sequencing platforms. The 16S rRNA gene assignments were annotated to (1) reinforce the taxonomic classifications based on six million significant protein assignments, (2) classify bacterial taxa to genus level, and (3) generate phylogenetic distances between samples. Phylogenetic distances between microbial communities were quantified using weighted Unifrac distances[62] derived from relative abundances of 16S Operational Taxonomic Units (clustered at the 97% sequence identity level via OptiClust[63]) because 16S genes can be differentiated using a quantitative genetic distance as opposed to protein classifications assigned to taxa of a particular rank. Statistical analyses were completed in R using the Vegan Package[57] and with JMP Pro v13 (SAS, Cary, NC, USA).

**Reporting summary**. Further information on experimental design is available in the Nature Research Reporting Summary linked to this article.

## Data availability

The biochemical (e.g., DOC, oxygen, pH) data that support the findings of this study have been deposited in [BCO-DMO (https://www.bco-dmo.org/project/675025)] under the dataset collection Line_Island_Diel_Tents and the metagenomic sequence data has been deposited into the [SRA] under accession codes SAMN10442328-SAMN10442375 with the following project code [NCBI, https://www.ncbi.nlm.nih.gov/bioproject/504905]. The source data underlying Figs. 2, 3, Supplementary Fig. 2 and Supplementary Tables 2, 3 are provided as a Source Data file.

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

## Acknowledgements

The diel survey was executed during a research expedition to the southern Line Islands funded by the Moore Family Foundation, the SDSU Vice President of Research, Scripps Institution of Oceanography, and several private donors. We thank the Captain, Martin Graser, and crew of the M/Y Hans Explorer for logistical support and hospitality. This work was carried out under research permits from the Environment and Conservation Division of the Republic of Kiribati. Thank you to Liz Dinsdale for providing her MiSeq platform to sequence the microbial metagenomes. Mark Little was also a tremendous help to LWK with the sequencing. We also thank Heather Maughan for her valuable edits to the manuscript. This research was sponsored by the GBMF Investigator Award 3781 (to F.R.) and the US National Science Foundation (awards OCE-1538567 to L.W.K., OCE-1538393 to C.E.N., OCE-1538428 to C.A.C. and PIRE-1243541 to F.R.). This paper is funded in part by a grant/cooperative agreement from the National Oceanic and Atmospheric Administration, Project A/AS-1, which is sponsored by the University of

Hawai'i Sea Grant College Program, SOEST, under Institutional Grant no. NA14OAR4170071 from NOAA Office of Sea Grant, Department of Commerce. The views expressed herein are those of the authors and do not necessarily reflect the views of NOAA or any of its subagencies. This is publication number 10569 of the School of Ocean and Earth Science and Technology of the University of Hawai'i at Mānoa and publication UNIHI-SEAGRANT-JC-16-25 of the University of Hawai'i Sea Grant College Program.

## Author contributions

L.W.K., C.E.N., and A.F.H. co-led the manuscript effort and contributed equally. L.W.K. led the sequencing and bioinformatic efforts. C.E.N. conducted the phylogenetic and statistical analyses. A.F.H., M.H., and F.R. designed and built the novel night sampling scheme. J.E.S., M.D.F., A.F.H., M.D.J., E.L.A.K., Y.W.L., R.A.E., M.J.A.V., B.Z., and F.R. designed the field study and collected all of the samples in situ. S.A.S., J.E.S., and F.R. provided funding for the research cruise. A.F.H., C.A.C., S.C., M.D.F., M.D.J., Z.A.Q., and J.E.S. analyzed the biogeochemical data. D.S.N., S.M., G.G.Z.S., and R.A.E. provided bioinformatic support. L.W.K., C.E.N., and A.F.H. analyzed the metagenomic sequence data and wrote the paper.

## Additional information

**Competing interests:** The authors declare no competing interests.

