## [Peer Review File · Nature Communications]

Reviewers' Comments:

Reviewer #1:

Remarks to the Author:

This manuscript by Wegley Kelly and colleagues used metagenomic sequencing of coral reef seawater to explore diurnal variation in the functional potential of the microbial population, indicating a dramatic shift between day and night that was primarily driven by a reduced relative abundance of *Psychrobacter* at night. This is an interesting and timely article that brings to light the importance of scale (in this case diurnal scale) when interpreting microbial data in the context of ecosystem dynamics.

The Introduction does a beautiful job of establishing context and positioning the research. The background information regarding microbial processes on coral reefs is introduced in a logical way and the relevance for the field of coral reef ecology is clear; i.e. the need to consider bias towards daytime sampling when benthic reef taxa have highest production and calcification.

My primary concern with this manuscript, and it is a very significant one, is the sampling strategy. The Authors primary aim which was so well set out in the introduction was to establish diurnal variation in microbial functional potential and community dynamics. Despite this aim, most of the replication occurs at the level of mesocosm and reef site. I cannot begin to understand why the Authors would have set up the sampling regime such that there is only a single night time sampling to compare with the 2 x daytime sampling events for each mesocosm? There is absolutely no power in such a design (essentially $n=1$) to test diurnal effects? While I appreciate that results were consistent across mesocosms, this is not sufficient for having statistical confidence in any day vs night comparison. The sequencing of 48 metagenomes is a substantial effort and the conclusions would have been much more compelling if this sequencing effort was directed to a minimum of 3 x day and 3 x night samplings for each replicate system.

Another concern pertaining to the methods regards the environmental relevance of performing the sampling in closed mesocosm systems, with no comparison to results obtained from an open reef system? The Authors therefore need to fully acknowledge this in the manuscript and make clear that these findings would need to be validated in an open reef environment (particularly because the pattern reported here is in contrast to the pattern previously reported in open water systems!) An analysis comparing the composition of the community in the closed mesocosms to open reef systems could be easily performed using previously published datasets for the Line islands by the same Authors. The abstract (particularly the last 2 sentences) should also be toned down accordingly.

While the metagenomics analysis is sound and the presentation of the data is nice, I am surprised that the Authors made no attempt to undertake a population genomic approach with this dataset? It would appear that no binning was performed? Considering the high abundance of specific taxa (such as the *Psychrobacter*), it should have been straightforward to obtain good quality genome bins which would have greatly strengthened the results?

Reviewer #2:

Remarks to the Author:

Overview:

The manuscript titled 'Diel population and functional synchrony of microbial communities on coral

reefs' written by Linda Wegley Kelly et al. presents a metagenomic analysis of reef water bacterial communities from the Southern Line Islands in the Central Pacific, which feature some of the most pristine coral reefs in the world. The authors find that reef bacterioplankton communities exhibit compositional shifts between day and night, which correspond to changes in metabolic profiles. The authors thereby provide insight into diel patterns of bacterial metabolism on coral reefs, with potential implications for the channeling of energy through the microbial loop (not empirically assessed in this study).

Overall, the paper is well written, the methodology is sound, and the work provides unprecedented first insights into diel taxonomic and functional profiles of coral reef bacterioplankton communities (previous studies were only conducted during day time). It should be noted that the here presented work has a limited temporal and geographic resolution (one single sampling campaign over 24 h per site, for 4 sites along a small remote island chain in the Central Pacific). As such, future efforts should target a larger geographic area (such as what is published in Haas et al. 2016 Nat Microbiol) to see if this is a universal observation.

The observed reported diel functional shifts (e.g., from anabolic to catabolic processes) are highly interesting, but also methodology-wise somewhat expected considering the dramatic changes in abundance of very few bacterial taxa (e.g. *Prochlorococcus*, *Psychrobacter*, and *Rhodobacteraceae*). Therefore, complementary approaches (e.g. transcriptomics) would be warranted to support the conclusions and to establish a functional link (for instance, see Cardenas et al. 2017 ISMEJ).

Finally, while I recognize the effort of characterizing bacterial communities using WGS over 16S rRNA amplicon sequencing to reduce bias associated with PCR, the low sequencing depth of this approach needs to be discussed and it needs to be pointed out how it would affect the study purpose. In particular, with regard to diversity data: the authors were able to predict ~25 thousand SSU rRNAs which represent ~520 reads per sample. This seems rather low to unequivocally support the conclusions.

General comments:

The authors need to clarify the sampling design in the methodology section, which in part become only apparent once the reader teases apart figures and results (details for time points, exact sample size and how numbers of replicates distribute over time points; $n = 16$ is stated in line 156, but this doesn't quite add up when comparing the data points in the MDS plot (Fig. S1A); line 134-135 states a total of 48 samples (i.e., samples distribute over 4 sites x 4 mesocosms x 3 sampling points?).

Geographic variation and diel cycles accounted for most of the taxonomic and functional variability between samples. Results clearly demonstrate a diel oscillation of *Psychrobacter* sp., *Prochlorococcus* sp., and *Rhodobacteraceae*. While this phenomenon has never been observed on coral reefs, previous findings address similar patterns driven by photosynthesis-derived DOC: (Ottesen EA, Young CR, Gifford SM, Eppley JM, Marin R, Schuster SC, Scholin CA, DeLong EF. Multispecies diel transcriptional oscillations in open ocean heterotrophic bacterial assemblages. *Science*. 2014 Jul 11;345(6193):207-12). It is well established that *Rhodobacteraceae* are efficient "DOC-eaters" that normally correlate negatively with DOC concentrations due their high repertoire to catabolize carbohydrates, while *Prochlorococcus* is a common member of the photosynthetic picoplankton community. These premises were corroborated by the authors through the analysis of functional profiles that showed higher abundance of carbohydrate catabolism genes at night and photosynthesis genes during the day.

One important aspect that I would like to see discussed: the observed strong diel shifts in bacterial diversity are surprising considering the short incubation times. Did the authors consider evaluating the

effects of confinement? Such drastic changes in the bacterial communities could eventually reflect magnification as result of bottle effects.

Specific comments:

Introduction

L75: I would not refer to "Coral reef ecosystems" but rather to the "benthic community"

L82-91: The link between bacterial demand and energy available for higher trophic levels is not clear to me. Isn't greater bacterial biomass (part of the particulate organic carbon pool) better to support microeukaryotic life and subsequently higher trophic levels (your very own explanation in L100-L102) rather than higher DOC availability? I understand from the Haas et al. 2016 Nat Microbiol microbialization paper that energy is used inefficiently to promote bacterial production but the whole point of this is that this energy is retained in basal trophic levels due to the lack of "intermediate predators" (?). This aspect should be included.

L85: Replacing "system" by "state" might be more appropriate

Methods

L136-138: Could you please clarify this part. I presume you ran a similarity search using RAPSearch against the SEED database implemented in SUPER-FOCUS and the best hits were used to do functional and taxonomical annotation of your reads? Can you comment how this is related to reference 30?

Results

I miss a plot showing the overall functional/taxonomic changes in the bacterial community (e.g. bar plots). Shifts of particular taxa are appropriate and better represented in tables and heatmaps, however, general patterns of bacterial diversity are often hidden or difficult to extract from those. Showing overall bacterial diversity will help people to understand and give better context of the data.

Line 185: typo - 'Synechococcus'

Discussion

Line 295: 'Environmental Sample Processor'. This term occurs in the manuscript only in this one specific location, and I had to google it to find out that it refers to the mesocosm setup. Please introduce earlier in the manuscript (at the end of the introduction, or in the methods) if you bring it up in the final 'conclusion' paragraph. Alternatively, omit this term altogether.

Reviewer #1 (Remarks to the Author):

This manuscript by Wegley Kelly and colleagues used metagenomic sequencing of coral reef seawater to explore diurnal variation in the functional potential of the microbial population, indicating a dramatic shift between day and night that was primarily driven by a reduced relative abundance of *Psychrobacter* at night. This is an interesting and timely article that brings to light the importance of scale (in this case diurnal scale) when interpreting microbial data in the context of ecosystem dynamics.

The Introduction does a beautiful job of establishing context and positioning the research. The background information regarding microbial processes on coral reefs is introduced in a logical way and the relevance for the field of coral reef ecology is clear; i.e. the need to consider bias towards daytime sampling when benthic reef taxa have highest production and calcification.

Thank you for your supportive comments!

My primary concern with this manuscript, and it is a very significant one, is the sampling strategy. The Authors primary aim which was so well set out in the introduction was to establish diurnal variation in microbial functional potential and community dynamics. Despite this aim, most of the replication occurs at the level of mesocosm and reef site. I cannot begin to understand why the Authors would have set up the sampling regime such that there is only a single night time sampling to compare with the 2 x daytime sampling events for each mesocosm? There is absolutely no power in such a design (essentially $n=1$) to test diurnal effects? While I appreciate that results were consistent across mesocosms, this is not sufficient for having statistical confidence in any day vs night comparison. The sequencing of 48 metagenomes is a substantial effort and the conclusions would have been much more compelling if this sequencing effort was directed to a minimum of 3 x day and 3 x night samplings for each replicate system.

The suggestion of the reviewer is that without three successive days and nights within the same deployment there can be no statistical power to resolve a diel pattern. The reviewer is emphasizing a key statistical issue: the issue of a lack of replication (i.e., “essentially $n=1$ ”). Our stated goal is to measure how day and night microbial communities differ. Studies of diel patterns in situ must grapple with prioritizing replication in time (i.e., multiple day-night cycles) or space (multiple locations or communities), with the ideal sampling strategy including both: capturing sequential day-night cycles in multiple distinct communities. The biggest problem with the former is the issue of pseudoreplication, with a spatial $n=1$. The biggest problem with the latter is the issue of time series, with a temporal $n=1$. Given the fundamental logistical constraint of our goal of ship-based sampling in remote reefs unimpacted by humans, we chose the latter priority to maximize spatial replication while minimizing confinement effects; we want to emphasize that this approach is at least as statistically robust as many of the shining examples in the recent literature choosing the former priority by quantifying diel cycling in microbial communities by tracking a single community through multiple day-night cycles (Ottensen et al., 2014; Aylward et al., 2015). We feel that our approach, deploying 4 parallel enclosures on 4 different islands over a period of 1 month is the most robust statistical design to resolve the degree to which day and night microbial communities differentiate and cycle, including both parallel and sequential spatial replication. There is no question that additional days and nights within enclosures exhibiting the same cycling would strengthen the pattern, but given the logistical constraints of sampling these chambers on remote reefs and the likelihood of artifacts arising from long term enclosure, we designed our sampling strategy to emphasize spatial and temporal replication as the strongest possible statistical argument for consistent changes between day and night in enclosed microbial communities: assessing patterns in multiple experimental enclosures on multiple islands over the shortest possible duration necessary to detect the pattern.

To clarify the statistical rigor built into our deployment and sampling design, the text pertaining to the experimental design has been changed to more clearly illustrate that samples were collected on four distinct reef sites over 24 hours, representing 1 ambient day sample (t0), 1 night sample (t12), and 1 day sample in the benthic chamber (t24), across four successive deployments (different dates) on four different island reef systems. Figure 2D is also a good depiction of the consistent diel patterns in community structure observed across all reef sites.

Lines 103-106: “The study site encompasses three uninhabited coral islands and one atoll: Vostok, Millennium, Starbuck, and Malden separated by 800km of latitudinal distance and exhibiting variance in ocean productivity and nutrient regimes between islands (Fox et al., 2018). On each of the four reef systems, seawater overlying the benthos (< 0.5m from the bottom) was collected from four distinct forereef sites over a 24 hour period (N=16).”

Lines 114-119: “Discrete water samples were collected from each reef plot (N=16) over a 24-hour diel cycle: between 0900-1159 on day 1 (t0), 2100-2359 on day 1 (t12) and 0900-1159 on day 2 (t24) for a total of 48 samples. On each of the four islands, samples were collected at one reef site without a benthic chamber at t12 and t24 to represent an open reef community without confinement. Time-zero samples were collected immediately after benthic chambers were deployed and therefore should be representative of the ambient reef water community.”

Finally, we strongly agree with the reviewers point that capturing a combination of sequential diurnal cycling in multiple locations would provide the most robust illustration of this phenomenon. A study without enclosures and based on land would be less constrained by time and three continuous day and night sample sets could be collected.

To address this point, Lines 293-305 have been added to the discussion: “The use of commercial autonomous samplers could provide the capacity to sample at finer temporal resolution (e.g., hourly) and over consecutive diel cycles. This enhanced time resolution would also provide a means to further characterize dynamics of microbial populations and help resolve the roles of predators, both phage and protists, in community shifts on diel time scales. While it is less logistically feasible to deploy larger sampling equipment on cruises, land-based field studies both on populated islands and remote regions (e.g., Palmyra; the northwest Hawaiian Islands) would provide greater geographic context to these observed diel patterns and allow for comparison across intact and degraded coral reef ecosystems. We hypothesize that the ecological functions driving these microbial fluxes will be diminished as habitats become influenced by anthropogenic perturbations. Further studies are required to better describe the retention of microbes from the water column into the benthic habitat, how this capacity serves to influence production and respiration processes in coral reef ecosystems, and to what degree this functionality is lost as habitats degrade.”

Another concern pertaining to the methods regards the environmental relevance of performing the sampling in closed mesocosm systems, with no comparison to results obtained from an open reef system? The Authors therefore need to fully acknowledge this in the manuscript and make clear that these findings would need to be validated in an open reef environment (particularly because the pattern reported here is in contrast to the pattern previously reported in open water systems!) An analysis comparing the composition of the community in the closed mesocosms to open reef systems could be easily performed using previously published datasets for the Line islands by the same Authors. The abstract (particularly the last 2 sentences) should also be toned down accordingly.

The manuscript has been corrected through-out to clarify that the in situ chambers were not closed systems, but experienced significant flushing through the reef matrix with a calculated water exchange rate of less than one hour.

1) “mesocosm” was removed from the entire manuscript because the term was used incorrectly. Since no experimental manipulation (e.g., temperature increase or nutrient enrichment) was imposed in this study, referring to the benthic chambers as mesocosms was inappropriate and has been corrected. We now refer to them as benthic enclosures.

2) We have added new Fig S1 to show fluorescein tracer measurements used to estimate water exchange within the benthic chambers in the supplementary methods:
“The rate of water exchange in the benthic chambers was calculated using dilution rates of Fluorescein dye. Fluorescein dye was injected into control chambers on Malden, Millennium, and Starbuck and the concentrations measured using a multiprobe sensor with a fluorometer (precision 0.01ppb; Manta2, Eureka Water Probes, TX, USA). Chamber flushing rates, F (liters min^{-1}) were calculated from the dilution rate, D (slope, $\log_{10}\text{ppb min}^{-1}$ divided by the initial concentration, $\log_{10}\text{ppb}$) multiplied by chamber volume (100 liters). Mean flushing rates were 5.04, 5.48, and 2.52 liters min^{-1} for Malden ($n=3$), Millennium ($n=4$), and Starbuck ($n=5$), respectively (Figure S1).”

We have also included a statement into the main text methods to clarify the capacity of confinement.

Lines 107-109-: “The forereef sites were subjected to high flushing through the reef matrix, therefore water exchange within the benthic chambers ranged between 2.52 and 5.48 liters min^{-1} (Figure S1 and Supplementary Methods).”

3) Supporting evidence that these community fluxes are representative of “open water systems” include:

- a) Community similarity between t_0 and t_{24} samples at the same site. Since the t_0 sample was collected immediately after deployment of benthic chambers, this sample is representative of ambient community structure. Figure S2a has been adapted to further illustrate the similarity between t_0 and t_{24} , which provides supporting evidence that the diel changes in community structure are not a direct effect of confinement.
- b) Samples from one open site, no benthic chamber, were collected on each island. The night community from one site on all four islands and one Vostok site at t_{24} was added to the MDS plot in Figure S2.
- c) Figure S3 summarizes the abundance of *Psychrobacter* spp. on open reefs collected during the daytime, both from our previously published datasets in the Line Islands (Fig S2b) and from eleven Hawaiian Islands collected in 2016.

4) Finally, we added a statements through-out the text highlighting supportive evidence that communities collected from the benthic chambers are representative of open systems; e.g., Lines 176-181: “Within each benthic chamber (reef site), microbial community phylogenetic and metagenomic composition differed more between night and day than between the daytime start (t_0) and end (t_{24}) of deployments, or between daytime samples among chambers ($p < 0.0001$; Figure S2B), suggesting both that daytime samples after 24h were representative of the ambient reef and that patterns in community structure were driven by temporal influences, rather than confinement effects of the benthic chambers.”

While the metagenomics analysis is sound and the presentation of the data is nice, I am surprised that the Authors made no attempt to undertake a population genomic approach with this dataset? It would appear that no binning was performed? Considering the high abundance of specific taxa (such as the

Psychrobacter), it should have been straightforward to obtain good quality genome bins which would have greatly strengthened the results?

This is a great suggestion and sequence assemblies were performed. However, obtaining good quality genome bins required combining the sequence libraries from several sites. Ultimately, we chose to analyze the data from unassembled sequence reads to preserve the site replicates at each time point. Future work will seek to use the combined datasets to investigate the population genomics of the most abundant organisms.

Reviewer #2 (Remarks to the Author):

Overview:

The manuscript titled ‘Diel population and functional synchrony of microbial communities on coral reefs’ written by Linda Wegley Kelly et al. presents a metagenomic analysis of reef water bacterial communities from the Southern Line Islands in the Central Pacific, which feature some of the most pristine coral reefs in the world. The authors find that reef bacterioplankton communities exhibit compositional shifts between day and night, which correspond to changes in metabolic profiles. The authors thereby provide insight into diel patterns of bacterial metabolism on coral reefs, with potential implications for the channeling of energy through the microbial loop (not empirically assessed in this study).

Overall, the paper is well written, the methodology is sound, and the work provides unprecedented first insights into diel taxonomic and functional profiles of coral reef bacterioplankton communities (previous studies were only conducted during day time). It should be noted that the here presented work has a limited temporal and geographic resolution (one single sampling campaign over 24 h per site, for 4 sites along a small remote island chain in the Central Pacific). As such, future efforts should target a larger geographic area (such as what is published in Haas et al. 2016 Nat Microbiol) to see if this is a universal observation.

We appreciate that the quality of the work is appreciated by the reviewer and agree completely that this work points immediately to the need for more widespread investigation of this phenomenon and how it varies in space and time among a broader diversity of reefs. We have added sentences to the discussion emphasizing these future directions explicitly,

Lines 292-305, as follows:

“The influence of predation versus rhythmic growth patterns on community structure are fundamental questions in microbial ecology that remain largely unanswered. The use of commercial autonomous samplers could provide the capacity to sample at finer temporal resolution (e.g., hourly) and over consecutive diel cycles. This enhanced time resolution would also provide a means to further characterize dynamics of microbial populations and help resolve the roles of predators, both phage and protists, in community shifts on diel time scales. While it is less logistically feasible to deploy larger sampling equipment on cruises, land-based field studies both on populated islands and remote regions (e.g., Palmyra; the northwest Hawaiian Islands) would provide greater geographic context to these observed diel patterns and allow for comparison across intact and degraded coral reef ecosystems. We hypothesize that the ecological functions driving these microbial fluxes will be diminished as habitats become influenced by anthropogenic perturbations. Further studies are required to better describe the retention of microbes from the water column into the benthic habitat, how this capacity serves to influence production and respiration processes in coral reef ecosystems, and to what degree this functionality is lost as habitats degrade.”

The observed reported diel functional shifts (e.g., from anabolic to catabolic processes) are highly interesting, but also methodology-wise somewhat expected considering the dramatic changes in abundance of very few bacterial taxa (e.g. Prochlorococcus, Psychrobacter, and Rhodobacteraceae). Therefore, complementary approaches (e.g. transcriptomics) would be warranted to support the conclusions and to establish a functional link (for instance, see Cardenas et al. 2017 ISMEJ).

Again, we agree wholeheartedly with the need to follow up this work with transcriptomic investigations to determine the degree to which shifts in population abundance (and corresponding community genomic content) relate to shifts in community transcriptomics. We have modified and streamlined the discussion to avoid overstating conclusions that warrant expression studies.

Finally, while I recognize the effort of characterizing bacterial communities using WGS over 16S rRNA amplicon sequencing to reduce bias associated with PCR, the low sequencing depth of this approach needs to be discussed and it needs to be pointed out how it would affect the study purpose. In particular, with regard to diversity data: the authors were able to predict ~25 thousand SSU rRNAs which represent ~520 reads per sample. This seems rather low to unequivocally support the conclusions.

We used a combined analysis to compare day-night microbial communities across whole genomes (6 million significant sequence assignments) classified as Family level taxons, then to identify taxa at a finer resolution, we extracted putative SSU rRNA genes and classified them “as ecotypes”. It is a strength of the data analysis that the metagenomic annotation (sequence similarities across whole genomes) is supported by the phylogenetic annotation (clustering and alignment) shown in Figure 2A-D and Figure S2B.

To further clarify this point in the text, we have added Lines: 176-181 “Within each benthic chamber (reef site), microbial community phylogenetic and metagenomic composition differed more between night and day than between the daytime start (t0) and end (t24) of deployments, or between daytime samples among chambers (p < 0.0001; Figure S2B), suggesting both that daytime samples after 24h were representative of the ambient reef and that patterns in community structure were driven by temporal influences, rather than confinement effects of the benthic chambers.”

General comments:

The authors need to clarify the sampling design in the methodology section, which in part become only apparent once the reader teases apart figures and results (details for time points, exact sample size and how numbers of replicates distribute over time points; n = 16 is stated in line 156, but this doesn't quite add up when comparing the data points in the MDS plot (Fig. S1A); line 134-135 states a total of 48 samples (i.e., samples distribute over 4 sites x 4 mesocosms x 3 sampling points?).

Text has been added to the Methods section to clarify the sampling methodology.

Lines 105-106: “On each island, seawater overlying the benthos (< 0.5m from the bottom) was collected from four distinct forereef sites over a 24 hour period (N=16).”

Lines 114-116: “Discrete water samples were collected from each reef plot (N=16) 3-times over a 24 hour diel cycle: between 0900-1159 on day 1 (t0), 2100-2359 on day 1 (t12) and 0900-1159 on day 2 (t24) for a total of 48 samples.”

Additional text has also been added to the Results section.

Lines 150-152: “The microbial community structure during the day (t0 and t24) and the night (t12) is described from four distinct reef sites on each of the four islands: Vostok, Millennium, Starbuck, and Malden (N=16 sites and 48 samples)”.

Geographic variation and diel cycles accounted for most of the taxonomic and functional variability between samples. Results clearly demonstrate a diel oscillation of *Psychrobacter* sp., *Prochlorococcus* sp., and Rhodobacteraceae. While this phenomenon has never been observed on coral reefs, previous findings address similar patterns driven by photosynthesis-derived DOC: (Ottesen EA, Young CR, Gifford SM, Eppley JM, Marin R, Schuster SC, Scholin CA, DeLong EF. Multispecies diel transcriptional oscillations in open ocean heterotrophic bacterial assemblages. *Science*. 2014 Jul 11;345(6193):207-12). It is well established that Rhodobacteraceae are efficient “DOC-eaters” that normally correlate negatively with DOC concentrations due their high repertoire to catabolize carbohydrates, while *Prochlorococcus* is a common member of the photosynthetic picoplankton community. These premises were corroborated by the authors through the analysis of functional profiles that showed higher abundance of carbohydrate catabolism genes at night and photosynthesis genes during the day.

One important aspect that I would like to see discussed: the observed strong diel shifts in bacterial diversity are surprising considering the short incubation times. Did the authors consider evaluating the effects of confinement? Such drastic changes in the bacterial communities could eventually reflect magnification as result of bottle effects.

We agree completely with the importance of ensuring that the observed patterns are not the result of confinement effects. We have now added explicit mention to our sampling design emphasizing that we used multiple deployments on multiple islands across multiple weeks explicitly to avoid confinement effects (Lines 105-109). The high similarity at each site between t0 samples (collected within minutes of the benthic chamber deployments) and the t24 samples provides significant evidence that the strong diel shifts are not the result of confinement (Fig 2E and Fig S2); this is now emphasized in Lines 176-181 (as mentioned above). Additionally, new data points from a subset of samples collected on open reefs (without a benthic chamber) have been added to Figure S2A.

We have also added a new Figure S1 to show fluorescein tracer measurements used to estimate water exchange within the benthic chambers.

New text has also been added to the Supplementary Methods: “The rate of water exchange in the benthic chambers was calculated using dilution rates of Fluorescein dye. Fluorescein dye was injected into control chambers on Malden, Millennium, and Starbuck and the concentrations measured using a multiprobe sensor with a fluorometer (precision 0.01ppb; Manta2, Eureka Water Probes, TX, USA). Chamber flushing rates, F (liters min^{-1}) were calculated from the dilution rate, D (slope, $\log_{10}\text{ppb min}^{-1}$ divided by the initial concentration, $\log_{10}\text{ppb}$) multiplied by chamber volume (100 liters). Mean flushing rates were 5.04, 5.48, and 2.52 liters min^{-1} for Malden ($n=3$), Millennium ($n=4$), and Starbuck ($n=5$), respectively (Figure S1).”

We have also included a statement into the maintext methods to clarify the capacity of confinement. Lines 105-109: “On each island, seawater overlying the benthos ($< 0.5\text{m}$ from the bottom) was collected from four distinct forereef sites over a 24-hour period ($N=16$). Benthic chambers were constructed over each reef plot (0.75m^2) to provide structure for autonomous sampling at night. The forereef sites were subjected to high flushing through the reef matrix, therefore water exchange within the benthic chambers ranged between 2.52 and 5.48 liters min^{-1} (Figure S1 and Supplementary Methods).”

Specific comments:

Introduction

L75: I would not refer to “Coral reef ecosystems” but rather to the “benthic community”

This has been changed.

L82-91: The link between bacterial demand and energy available for higher trophic levels is not clear to me. Isn't greater bacterial biomass (part of the particulate organic carbon pool) better to support microeukaryotic life and subsequently higher trophic levels (your very own explanation in L100-L102) rather than higher DOC availability? I understand from the Haas et al. 2016 Nat Microbiol microbialization paper that energy is used inefficiently to promote bacterial production but the whole point of this is that this energy is retained in basal trophic levels due to the lack of “intermediate predators” (?). This aspect should be included.

Thank you for catching the miscommunication here. This paragraph has been re-structured and the statement about energy availability to higher trophic levels has been removed.

L85: Replacing “system” by “state” might be more appropriate
“system” has been changed to “state”

Methods

L136-138: Could you please clarify this part. I presume you ran a similarity search using RAPSearch against the SEED database implemented in SUPER-FOCUS and the best hits were used to do functional and taxonomical annotation of your reads? Can you comment how this is related to reference 30?

Thank you for identifying the lack of clarity here.

Additional text has been added to the Methods section to better describe the sequence annotation pipeline.

Lines 127-133: “Sequence reads were compared to the SEED database (31) for metabolic and taxonomic assignments using SUPERFOCUS, which aligns sequence similarities using RAPSearch2 and performs a 98% clustering of the proteins in the database to reduce computational taxation (32). Metagenomic reads aligned against the SEED database to get the functional annotation were subsequently extracted to identify the taxa that are encoding the respective protein coding genes. The taxonomic lineages were extracted from NCBI using a bioinformatics tool Taxonkit (<http://bioinf.shenwei.me/taxonkit/>).”

Reference 30, Edwards et al., 2006, has been changed to Overbeek et al., 2005. The citation was included to reference the SEED hierarchical subsystem classification of gene functions.

Results

I miss a plot showing the overall functional/taxonomic changes in the bacterial community (e.g. bar plots). Shifts of particular taxa are appropriate and better represented in tables and heatmaps, however, general patterns of bacterial diversity are often hidden or difficult to extract from those. Showing overall bacterial diversity will help people to understand and give better context of the data.

Because the aim of the article was to highlight the taxonomic changes between day and night microbial communities, bar charts were presented for the significantly enriched taxonomic groups (Fig2A-B). A majority of the bacterial community is presented for each island (mean abundance at each reef site plus s.e.m.) during both the day (~70%) and the night (~60%).

The following text to the legend of Fig 2: “Bar plots illustrate a majority of the bacterial diversity during the day (mean = 73%) and night (mean = 57%). Rarer taxa significantly enriched in the night are listed in Table S2.”

Line 185: typo – ‘Synechococcus’

This has been fixed – thank you for catching it.

Discussion

Line 295: ‘Environmental Sample Processor’. This term occurs in the manuscript only in this one specific location, and I had to google it to find out that it refers to the mesocosm setup. Please introduce earlier in the manuscript (at the end of the introduction, or in the methods) if you bring it up in the final ‘conclusion’ paragraph. Alternatively, omit this term altogether.

This term has been omitted and the following text has added.

Lines 293-297: “The use of commercial autonomous samplers could provide the capacity to sample at finer temporal resolution (e.g., hourly) and over consecutive diel cycles. This enhanced time resolution would also provide a means to further characterize dynamics of microbial populations and help resolve the roles of predators, both phage and protists in community shifts on diel time scales.”

Reviewers' Comments:

Reviewer #1:

Remarks to the Author:

The revised manuscript by Wegley and colleagues is certainly improved, and their response to reviewers is well considered. Whilst I feel my concern over the optimal sampling strategy for testing diurnal effects still holds, I appreciate the logistical challenges associated with night-time reef sampling. The Authors have certainly analysed this in the best way possible for this particular sample set and have described future work / sampling designs that would help validate these preliminary findings. As mentioned in my previous review, this is an interesting and timely article, and I would very much like to see it published. However I still have significant concerns outlined below.

1) Please make the sequence data available for reviewing. It is not sufficient to state that the data is available under accession numbers XXXXX. Reviewers need to be able to access the data for robust reviewing.

2) The introduction sets the scene for why there may be microbial differences between day and night in the context of the benthic community structure (in fact 2 whole paragraphs are dedicated to this rationale (lines 67-88). Fig 1B also shows very large differences in community structure of the benthos, both between replicate tents within a site and also between sites. However, despite the rationale that differences in benthic production and calcification could result in variation in microbial functional potential, the manuscript does not test whether any of the microbial data reflects variation in these macro-ecological features? This statistical analysis should be performed and included in the main manuscript.

3) The authors should provide the genome (or genome bin) for the Psychrobacter that drives most of the community differences. This comprises a very large fraction of the community in some samples and would provide confidence that the differences in metabolic pathways are indeed driven by variation in the abundance of this organism. It would also enable the authors to construct a genome tree to better phylogenetically place this microorganism (see concern below about the origin of the Psychrobacter).

4) The overwhelming abundance of Psychrobacter is unexpected as this is reported as a psychrophilic microorganism affiliated with polar waters (the authors acknowledge this in their discussion). However, Psychrobacter has also previously been described from coral mucus, hence the authors should consider whether the abundance of this organism in the daytime samples may actually be an artefact of coral mucus being captured on the seawater filters during daytime sampling rather than this being a true component of the bacterioplankton (in the same way this could be a host-associated member of some zooplankton that only occupy the reef during daytime)? The methodological approach described for seawater filtering would not eliminate larger zooplankton, mucus, marine snow and other detritus, as a single 0.2 um filtering step is used, and it is possible that these factors vary diurnally.

5) I would like to see some comparison (beyond the Psychrobacter) for how the community profiles described here compare with previous studies of the same (or similar) reefs. Many of the results are surprising – the comparatively low abundance of SAR11 as well as the presence of taxa such as Neisseriaceae and Pasturellaceae which were significantly enriched in daytime sampling – these are generally considered commensal microorganisms of mammals and other eukaryotes, indicating likely contamination of the seawater microbiomes.

6) The methodological details need revising so that the reader can understand the sampling and

statistical designs. For instance, the authors state that the 1 x non-enclosed sampling effort / site was conducted at T=12h and T=24h to address any effect of enclosure. However these samples are not reflected in the NMDS ordination Figure S2A and no statistical results are presented for this comparison? In fact, Fig S2A raises more questions as the non-enclosed samples actually cluster quite separately from the enclosed samples (particularly for Vostock and Millenium sites), indicating quite a dramatic effect of 'enclosure'?

Reviewer #2:

Remarks to the Author:

The authors adequately addressed my concerns, except for the comment with regard to the ~520 SSU rRNA reads per sample. The authors explain how their approach still yields significant results, which I don't doubt. However, the low sequencing depth should be explicitly stated in the manuscript.

Reviewers' comments:

Reviewer #1 (Remarks to the Author):

The revised manuscript by Wegley and colleagues is certainly improved, and their response to reviewers is well considered. Whilst I feel my concern over the optimal sampling strategy for testing diurnal effects still holds, I appreciate the logistical challenges associated with night-time reef sampling. The Authors have certainly analysed this in the best way possible for this particular sample set and have described future work / sampling designs that would help validate these preliminary findings. As mentioned in my previous review, this is an interesting and timely article, and I would very much like to see it published. However I still have significant concerns outlined below.

We greatly appreciate the time this reviewer has put forth evaluating this study. The critiques are well founded, insightful, and have helped strengthen the manuscript considerably.

1) Please make the sequence data available for reviewing. It is not sufficient to state that the data is available under accession numbers XXXXX. Reviewers need to be able to access the data for robust reviewing.

The sequence data has been made public through NCBI and the following text has been added to Lines 332-335: “The biochemical (e.g., DOC, oxygen, pH) data that support the findings of this study have been deposited in [BCO-DMO (<https://www.bco-dmo.org/project/675025>)] and the metagenomic sequence data has been deposited into the [SRA] with the following project code [NCBI, <http://www.ncbi.nlm.nih.gov/bioproject/504905>].”

2) The introduction sets the scene for why there may be microbial differences between day and night in the context of the benthic community structure (in fact 2 whole paragraphs are dedicated to this rationale (lines 67-88). Fig 1B also shows very large differences in community structure of the benthos, both between replicate tents within a site and also between sites. However, despite the rationale that differences in benthic production and calcification could result in variation in microbial functional potential, the manuscript does not test whether any of the microbial data reflects variation in these macro-ecological features? This statistical analysis should be performed and included in the main manuscript.

We certainly agree that incorporating benthic composition as a factor in multivariate models may help explain additional variance in microbial community structure among mesocosms. Our previous publication on these islands explored this concept in detail (Kelly et al., 2014, reference 14). We have added statistical tests (PERMANOVA) to the Results section addressing island level variation to Lines 181-183 as follows: “Microbial communities differed significantly by Island across all samples and within both day and nighttime subsets of the data (PERMANOVA $p < 0.001$), emphasizing the role of island-scale variation in benthic communities (Figure 1B, Table S1) in structuring microbial consortia (14).”

However, we feel that additional exploration of the statistical relationships between benthic community structure and microbial community structure would distract from the specific hypotheses tested in this study, and is unlikely to clarify our understanding of the system because we did not set out to capture the breadth of benthic community composition as we have done in previous work (REFs: Kelly et al., 2014; Haas et al., 2016). In line with our goal to present a refined story to a broad scientific audience illustrating consistent diel shifts in functional characteristics inherent among relatively pristine reef ecosystems, we hope the reviewer can recognize that we have made a conscious effort to streamline the manuscript and avoid complicating the central thesis. While there are some community differences observed across islands (e.g., on Starbuck, which has a higher percent cover of calcified macroalgae), the reef function that elicits the observed day-night community fluctuations was conserved; we aimed to highlight an ecological process inherent to these remote, calcifier-dominated reef systems.

3) The authors should provide the genome (or genome bin) for the Psychrobacter that drives most of the community differences. This comprises a very large fraction of the community in some samples and would provide confidence that the differences in metabolic pathways are indeed driven by variation in the abundance of this organism. It would also enable the authors to construct a genome tree to better phylogenetically place this microorganism (see concern below about the origin of the Psychrobacter).

The prevalence of Psychrobacter in these reef water communities is intriguing. We agree with the reviewer that the assembly of the Psychrobacter genome from the shotgun sequence reads would provide a greater understanding of the contribution of the Psychrobacter to the day-night variation in community metabolism. We have initiated assembly of the reef derived Psychrobacter genome. However a robust genome assembly from a mixed population is by no means trivial and we feel strongly that this approach is beyond the scope of this article.

To address the reviewers comment and ensure we do not directly attribute the daytime community metabolism to the relative abundance of unassembled sequence similarities to Psychrobacter, the following statement has been removed from the discussion: “The metabolic profile during the day was strongly influenced by the high prevalence of *Psychrobacter* spp.”

4) The overwhelming abundance of Psychrobacter is unexpected as this is reported as a psychrophilic microorganism affiliated with polar waters (the authors acknowledge this in their discussion). However, Psychrobacter has also previously been described from coral mucus, hence the authors should consider whether the abundance of this organism in the daytime samples may actually be an artefact of coral mucus being captured on the seawater filters during daytime sampling rather than this being a true component of the bacterioplankton (in the same way this could be a host-associated member of some zooplankton that only occupy the reef during daytime)? The methodological approach described for seawater filtering would not eliminate larger zooplankton, mucus, marine snow and other detritus, as a single 0.2 um filtering step is used, and it is possible that these factors vary diurnally.

Regarding the phylogenetic placement and origin of the *Psychrobacter* spp. we have completed further analyses and added new Figure S3 to demonstrate that the putative reef *Psychrobacter* do indeed cluster between numerous isolates designated to be from the genus *Psychrobacter*.

We have also added the following text to the discussion, Lines 266-270: “The genus *Psychrobacter* has been characterized as aerobic, non-motile, cold-adapted and mesophilic (55-56). Designating the most abundant taxon inhabiting coral reef waters as psychrophilic was surprising. Hence, this discovery that a closely related population of *Psychrobacter* spp. (Figure S3) can dominate tropical marine microbial communities warrants further investigation to better understand the ecology of this clade.”

Figure S3. Alignment of shotgun metagenomic 16S rRNA coding sequences assigned to OTUs classified as *Psychrobacter* by SINA alignment to the SILVA database. Distribution density (A), alignment length (B) and alignment identity (C) of reads clustered into positional OTUs (D) and consensus sequences visualized in a maximum likelihood phylogeny (E) with nearest neighbors from the SILVA v132 RefNR database and built by RaxML using the GTR model.

Coral mucus associated organisms would not be hypothesized to exhibit diel variation as there is no evidence that mucus production varies between day and night. Further, our sampling protocol would not allow for the distinction between particle-associated and free living members of the bacterial community. Text has been included in the discussion to address the requirement of further studies to better characterize this mechanism. Lines 308-314: “The influence of predation versus rhythmic growth patterns on community structure are fundamental questions in microbial ecology that remain largely unanswered. The use of commercial autonomous samplers could provide the capacity to sample different size fractions (i.e., particle associated versus free-living cells), at finer temporal resolution (e.g., hourly) and over consecutive diel cycles. This enhanced time resolution would also provide a means to further characterize dynamics of microbial populations and help resolve the roles of predators, both phage and protists, in community shifts on diel time scales.”

5) I would like to see some comparison (beyond the Psychrobacter) for how the community profiles described here compare with previous studies of the same (or similar) reefs. Many of the results are surprising – the comparatively low abundance of SAR11 as well as the presence of taxa such as Neisseriaceae and Pasteurellaceae which were significantly enriched in daytime sampling – these are generally considered commensal microorganisms of mammals and other eukaryotes, indicating likely contamination of the seawater microbiomes.

For the same reasons mentioned in our response to Comment 2, we do not seek to justify the presence of every bacterial lineage in this diverse marine ecosystem. We see no indication of contamination, and it is common to find abundant sequences belonging to a variety of lineages in environmental samples that were previously associated only with host associated habitats; sequences in the Neisseriaceae and Pasteurellaceae have been recently observed to be abundant in other reef systems (PMID: 23936086 and PMID: 22283330, respectively). The stability of reef microbial community structure during the day combined with the consistent community shifts at night across time and space provide further confidence that seawater microbiomes were not contaminated as all of the samples were extracted and sequenced simultaneously.

6) The methodological details need revising so that the reader can understand the sampling and statistical designs. For instance, the authors state that the 1 x non-enclosed sampling effort / site was conducted at T=12h and T=24h to address any effect of enclosure. However these samples are not reflected in the NMDS ordination Figure S2A and no statistical results are presented for this comparison? In fact, Fig S2A raises more questions as the non-enclosed samples actually cluster quite separately from the enclosed samples (particularly for Vostock and Millennium sites), indicating quite a dramatic effect of ‘enclosure’?

We have included new statistical analyses to directly compare the t0 samples (representing ambient daytime reef samples) to t24 samples (daytime following enclosure for 24 hours) as well as ambient nighttime reef samples (one per island) to t12 samples (nighttime following enclosure for 12 hours) to Lines 181-189 as follows: “Microbial communities differed significantly by Island across all samples and within both day and nighttime subsets of the data (PERMANOVA $p < 0.001$), emphasizing the role of island-scale variation in benthic

communities (Figure 1B, Table S1) in structuring microbial consortia (14). Across all islands microbial communities did not differ statistically between ambient (t0) and chamber endpoint (t24) daytime communities (PERMANOVA $p > 0.25$) or between ambient and chamber nighttime (t12) communities (PERMANOVA $p > 0.37$) but both ambient and chamber diel pairwise comparisons were all significantly different (PERMANOVA $p < 0.05$; Figure S2C). There was no significant interaction between Island and Time or between Island and inside vs. outside of chambers (PERMANOVA $p > 0.05$), clarifying that Islands did not differ in these robust patterns.”

For clarity, we have revised the following text in the Methods section. Lines 114-119: “Discrete water samples were collected from each reef plot (N=16) over a 24-hour diel cycle: ambient water between 0900 and 1159 on day 1 (t0), enclosure water between 2100 and 2359 on day 1 (t12) and enclosure water between 0900 and 1159 on day 2 (t24) for a total of 48 samples. On each of the four islands, samples were collected at one reef site without a benthic chamber between 2100 and 2359 on day 1 to represent an open reef nighttime community without confinement. Time-zero samples were collected immediately after benthic chambers were deployed and are considered representative of the ambient reef water community.” and Lines 123-126: “For metagenomic comparisons of the microbial community, 3 time points collected over a diel cycle from 4 reef sites on each of the 4 islands were sequenced and analyzed for a total 48 experimental samples plus 4 ambient nighttime samples, one collected from an adjacent open reef site on each island (depicted in Figure S2).”

Figure S2: Day-night community dissimilarity of reef microbes. (A) Nonmetric Multidimensional scaling (NMDS, stress = 0.13) of the metagenomic taxonomic composition of the reef microbial communities (Family level). Pairwise comparisons are grouped according to time of day (symbols) and location (categories across the x-axis); “Outside” refers to nighttime samples collected from the ambient reef, while T0 samples are considered daytime samples collected from the ambient reef. (B) Comparison of pairwise microbial community distances of protein coding genes (metagenomic dissimilarity) and species composition (phylogenetic dissimilarity) among sample sets; error bars depict the 95% confidence interval of the mean. (C)

PERMANOVA tests (package *adonis* in R) of pairwise community dissimilarity showing that daytime ambient (T0) and tent (T24) communities do not differ ($p = 0.57$), nighttime ambient (outside) and tent (T12) communities do not differ ($p = 0.57$), but all other pairwise comparisons are significant ($p < 0.05$). Including Island in the full model or either daytime or nighttime data subsets always yields a significant Island effect and a nonsignificant interaction term, clarifying that diel patterning does not differ among islands.

Reviewer #2 (Remarks to the Author):

The authors adequately addressed my concerns, except for the comment with regard to the ~520 SSU rRNA reads per sample. The authors explain how their approach still yields significant results, which I don't doubt. However, the low sequencing depth should be explicitly stated in the manuscript.

We thank this reviewer for their hard work and thoughtful guidance to substantially improve this manuscript. To address this final comment, we have added statement to the following text to Lines 142-146 in order to qualify the subpar sequence depth for the 16S rRNA gene reads.

“The average number of 16S reads per library was 528 (Table S1), a lower sequencing depth than amplicon libraries typically generated using next generation sequencing platforms. The 16S rRNA gene assignments were annotated to 1) reinforce the taxonomic classifications based on six million significant protein assignments, 2) classify bacterial taxa to genus level, and 3) generate phylogenetic distances between samples.”

Reviewers' Comments:

Reviewer #1:

Remarks to the Author:

I appreciate the significant effort the authors have put in to address the concerns I raised over 2 rounds of review. Their careful attention to these points have improved the manuscript and I am confident that it will be a robust and valuable contribution to the field.

REVIEWERS' COMMENTS:

Reviewer #1 (Remarks to the Author):

I appreciate the significant effort the authors have put in to address the concerns I raised over 2 rounds of review. Their careful attention to these points have improved the manuscript and I am confident that it will be a robust and valuable contribution to the field.

Thank you!